# A Vision-free baseline for Multimodal Grammar Induction

## Abstract

Past work has shown that paired vision-language signals substantially improve grammar induction in multimodal datasets such as MSCOCO. We investigate whether advancements in large language models (LLMs) that are only trained with text could provide strong assistance for grammar induction in multimodal settings. We find that our text-only approach, an LLM-based C-PCFG (LC-PCFG), outperforms previous multi-modal methods, and achieves state-of-the-art grammar induction performance for various multimodal datasets. Compared to image-aided grammar induction, LC-PCFG outperforms the prior state-of-the-art by 7.9 Corpus-F1 points, with an $85\%$ reduction in parameter count and $1.7\times$ faster training speed. Across three video-assisted grammar induction benchmarks, LC-PCFG outperforms prior state-of-the-art by up to 7.7 Corpus-F1, with $8.8\times$ faster training. These results shed light on the notion that text-only language models might include visually grounded cues that aid in grammar induction in multimodal contexts. Moreover, our results emphasize the importance of establishing a robust vision-free baseline when evaluating the benefit of multimodal approaches.

## 1 Introduction

Do vision-free models help unsupervised grammar induction in multimodal settings? Based on the semantic bootstrapping hypothesis (Pinker, 1984), recent work has tested whether pairing text with extralinguistic inputs during LLM training facilitates grammar acquisition. These studies found that performance on grammar induction improves when text training inputs are paired with other modalities, such as images (Zhao & Titov, 2020; Shi et al., 2019), videos (Zhang et al., 2022a), audio recordings (Zhang et al., 2021) or facial semantics (Zhang et al., 2021). These studies suggest that paired multimodal training data helps to ground text-only representations to the multimodal world and thereby facilitates grammar acquisition. However, there is limited direct evidence for the hypothesized importance of this grounding signal from other modalities into text. Prior studies of multimodal grammar induction compared multimodal training to weak text-only baselines. These text-only baselines used lexical word embeddings trained with little data or learnt from scratch, and the benefit found from paired multimodal training may simply be due to the total volume of data used to train the multimodal models. Also, multimodal training often requires demanding running time and processed pair-wised multimodal data which entails substantial pre-processing efforts. Therefore, we aim to design a strong vision-free baseline that leverages recent advances in LLMs, which potentially encompass visually grounded signals that aid in grammar induction in multimodal settings.

Large language models (LLMs) have revolutionized the field of natural language processing (Brown et al., 2020; Chowdhery et al., 2022) and by extension, multimodal vision-language learning (Radford et al., 2021; Ramesh et al., 2022). Recent results across domains suggest that LLMs often have a surprisingly detailed understanding of object-oriented concepts and physical mechanics of the world. For example, Imagen (Saharia et al., 2022) generates detailed images conditioned on textual prompt embeddings from a frozen T5 (Raffel et al., 2020) encoder, pre-trained on a purely text cloze task (Taylor, 1953). In robotics, SayCan (Ahn et al., 2022) demonstrates that LLMs contain rich high-level step planning information that can be readily exploited for a real-world robotic task when paired with a low-level actuator and controller. InstructGPT (Ouyang et al., 2022) and ChatGPT reveal the strong *apparent* grounding that can result from optimizing language models for human dialogue. Viewed under the lens of learning real-world concepts independent of the modality the

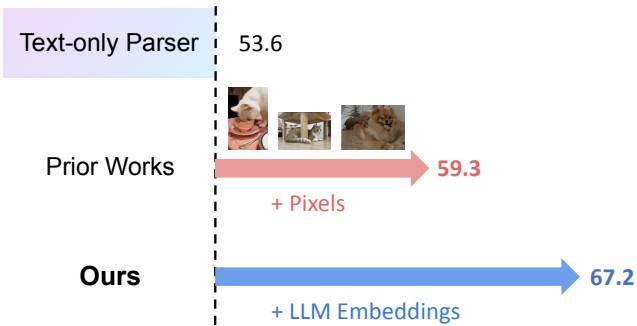

Figure 1: Comparison with prior multimodal methods on image-assisted grammar induction. Prior works suggested that paired images can *ground* sentences in pixels, thereby improving grammar induction (53.6 → 59.3 Corpus-level F1). We propose to instead build on text-only large-language models (LLM), and observe significant improvements in grammar induction (53.6 → 67.2).

concepts are expressed in, these results indicate that LLMs learn *meaning* far beyond what might be naïvely expected from a system trained for the cloze task.

In this work, we examine whether recent advancements in LLMs can help unsupervised constituency parsing. We find that LLM embeddings enable large improvements to text-only parsers, and substantially outperform results obtained from previous multimodal methods (Zhao & Titov, 2020). our code to inspire further research in this domain. Our key contributions are as follows:

**1.** Our LLM-based vision-free model (LC-PCFG) establishes a new state-of-the-art for image-based grammar induction. LC-PCFG outperforms prior works by at least 7.9 Corpus-F1 points, with an 85% reduction in trainable parameters and $1.7\times$ faster training speed.

**2.** LC-PCFG establishes a new state-of-the-art for video-based grammar induction, outperforming prior works across three video-based text parsing benchmarks, by up to 7.7 Corpus-F1 points and $8.8\times$ faster training speed.

**3.** We report and observe that our vision-free approach consistently outperforms other multimodal methods across various unsupervised validation criteria.

## 2 RELATED WORK

**Unsupervised parsing**   Unsupervised parsing, the task of inducing syntactic structure from text, has been extensively studied over the past few decades (e.g., Lari & Young, 1990; Carroll & Charniak, 1992; Clark, 2001; Klein & Manning, 2002; Smith & Eisner, 2005).

Many methods for unsupervised parsing rely on signals from text alone (e.g., Lari & Young, 1990; Carroll & Charniak, 1992; Klein & Manning, 2002; Shen et al., 2018; 2019). However, recent work suggested that multimodal signals may be needed for accurate grammar induction (Shi et al., 2019; Zhao & Titov, 2020; Zhang et al., 2021; 2022a). Prior work argued that the correspondence between visual features and certain words may facilitate the identification of syntactic constituents and therefore serve as a key component for unsupervised grammar induction (Shi et al., 2019). Subsequent studies built upon this intuition, showing that adding visual and auditory features to random word embeddings, or lexical word embeddings such as fastText, can improve model performance on unsupervised parsing tasks (Zhao & Titov, 2020; Shi et al., 2019; Zhang et al., 2022a; 2021).

**LLM features for grammar induction.**   Recent advances in large pretrained language models have enabled vast improvements on a wide range of downstream tasks, including syntactic parsing (e.g., Devlin et al., 2019; Radford et al., 2019; Kitaev et al., 2018). Prior works on the effect of grounding for unsupervised parsing have focused on text-only baselines based on much weaker lexical representations. Here we study whether the stronger word representations learned by current pretrained language models could contain equivalent multimodal grounded cues for unsupervised parsing.

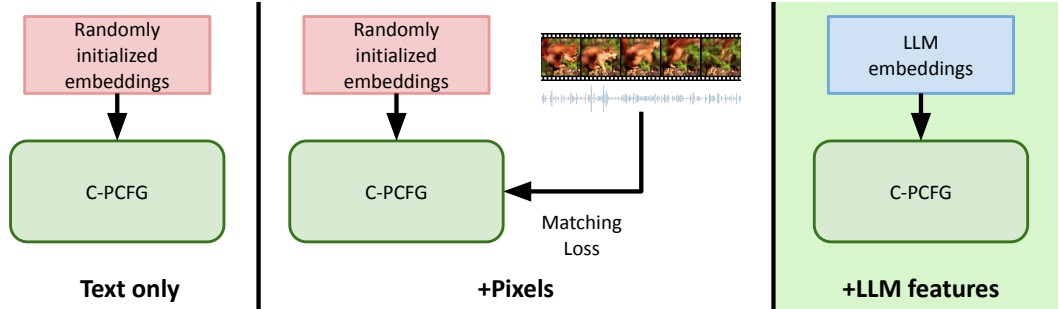

Figure 2: **Experimental Settings.** We explore using large language model features for unsupervised grammar induction. We use three experimental settings – (1) the standard setting in which word representations are learned from scratch (**Text Only**), (2) prior methods that incorporate a multimodal regularization loss (**+Pixels**), and (3) our method, which uses pre-trained text-only LLM features (**+LLM features**). We show that LLM features can obtain state-of-the-art performance, without requiring multimodal regularization.

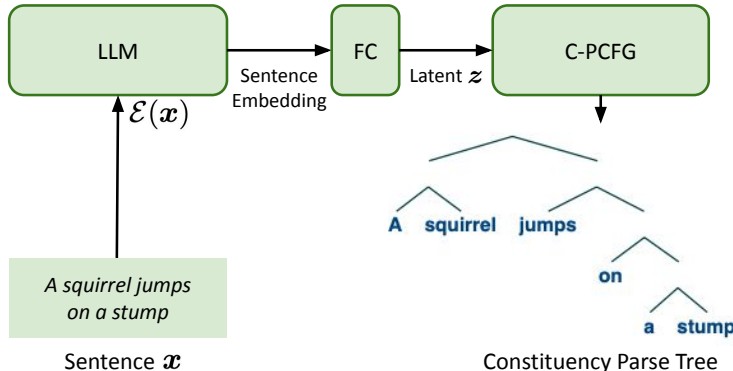

Figure 3: **LC-PCFG workflow.** A sentence $x$ is fed to an LLM to obtain a sentence embedding $\mathcal{E}(x)$. $\mathcal{E}(x)$ is passed through a fully-connected layer (FC), producing the latent $z$. $z$ is fed to the C-PCFG to obtain a constituency parse tree. Note that unlike prior work, our approach does not require multimodal data.

## 3   LC-PCFG: GRAMMAR INDUCTION WITH LARGE LANGUAGE MODELS

The goal of statistical grammar induction is to automatically induce syntactic structure over a text corpus. To do this, a grammar formalism is assumed, and the grammar parameters are optimized to fit the data. In this work, we investigate whether a vision-free baseline can surpass the performance of multimodal approaches trained on multimodal data for the task of unsupervised constituency parsing (See Figure 2 for an overview). We use Compound Probabilistic Context-Free Grammars (C-PCFGs) (Kim et al., 2019) as a testbed, constructing a C-PCFG that incorporates LLM representations (we refer to our model as LC-PCFG). We compare to prior methods that incorporated multimodal data (Zhao & Titov, 2020; Zhang et al., 2021; 2022a).

**Background.** C-PCFGs extend the Probabilistic Context Free Grammar (PCFG) formalism, and are defined by a 5-tuple $\mathcal{G} = (S, \mathcal{N}, \mathcal{P}, \Sigma, \mathcal{R})$, consisting of a start symbol $S$, a set of non-terminals $\mathcal{N}$, a set of pre-terminals $\mathcal{P}$, a set of terminals $\Sigma$, and a set of derivation rules $\mathcal{R}$:

$$\begin{aligned}
S &\to A & A &\in \mathcal{N} \\
A &\to BC & A &\in \mathcal{N}, B, C \in \mathcal{N} \cup \mathcal{P} \\
T &\to w & T &\in \mathcal{P}, w \in \Sigma
\end{aligned}$$

PCFGs define a probability distribution over transformation rules $\boldsymbol{\pi} = \{\pi_r\}_{r \in \mathcal{R}}$, and inference may be performed efficiently over them using the inside algorithm (Baker, 1979). In neural PCFGs, this

distribution may be formulated as follows:

$$\pi_{S \to A} = \frac{\exp(\boldsymbol{u}_A^\top f_1(\boldsymbol{w}_S))}{\Sigma_{A' \in \mathcal{N}} \exp(\boldsymbol{u}_{A'}^\top f_1(\boldsymbol{w}_S))}$$

$$\pi_{A \to BC} = \frac{\exp(\boldsymbol{u}_{BC}^\top \boldsymbol{w}_A)}{\Sigma_{B'C' \in \mathcal{M}} \exp(\boldsymbol{u}_{B'C'}^\top \boldsymbol{w}_A)}$$

$$\pi_{T \to w} = \frac{\exp(\boldsymbol{u}_w^\top f_2(\boldsymbol{w}_T))}{\Sigma_{w' \in \Sigma} \exp(\boldsymbol{u}_{w'}^\top f_2(\boldsymbol{w}_T))}$$

where $\boldsymbol{u}$ are transformation vectors for each production rule, $\boldsymbol{w}$ are learnable parameter vectors for each symbol, and $f_1$ and $f_2$ are neural networks. This formulation allows neural PCFGs to enjoy the benefits of fast inference as before, while additionally leveraging distributional representations through symbol embeddings.

The strong context-free assumption does not allow PCFGs to leverage global information when computing production probabilities in inference. Compound PCFGs (Kim et al., 2019) remedy this by formulating rule probabilities as a compound probability distribution (Robbins, 1956):

$$\boldsymbol{z} \sim p_\gamma(\boldsymbol{z}) \qquad\qquad \pi_{\boldsymbol{z}} = f_\lambda(\boldsymbol{z}, \boldsymbol{E}_{\mathcal{G}})$$

Where $\boldsymbol{z}$ is a latent variable generated by a prior distribution (generally assumed to be spherical Gaussian) and $\boldsymbol{E}_{\mathcal{G}} = \{\boldsymbol{w}_N | N \in \{S\} \cup \mathcal{N} \cup \mathcal{P}\}$ denotes the set of symbol embeddings. Rule probabilities $\boldsymbol{\pi}_{\boldsymbol{z}}$ are additionally conditioned on this latent:

$$\pi_{\boldsymbol{z}, S \to A} \propto \exp(\boldsymbol{u}_A^\top f_1([\boldsymbol{w}_S; \boldsymbol{z}])),$$

$$\pi_{\boldsymbol{z}, A \to BC} \propto \exp(\boldsymbol{u}_{BC}^\top [\boldsymbol{w}_A; \boldsymbol{z}]),$$

$$\pi_{\boldsymbol{z}, T \to w} \propto \exp(\boldsymbol{u}_w^\top f_2([\boldsymbol{w}_T; \boldsymbol{z}]))$$

Importantly, this formulation allows global information to be shared across production decisions during parsing through the latent $\boldsymbol{z}$ while maintaining the context-free assumption needed for efficient inference when $\boldsymbol{z}$ is fixed.

C-PCFGs are optimized with variational methods (Kingma & Welling, 2013), since the introduction of $\boldsymbol{z}$ makes inference intractable. At inference time, given a sentence $\boldsymbol{x}$, the variational inference network $q_\phi$ is used to produce the latent $\boldsymbol{z} = \boldsymbol{\mu}_\phi(g(\mathcal{E}(\boldsymbol{x})))$. Here, $g$ is a sentence encoder used to generate a vector representation given token embeddings $\mathcal{E}(\boldsymbol{x})$. For a more thorough treatment of C-PCFGs, please see Kim et al. (2019).

**LLM-based C-PCFG for grammar induction.** We design LC-PCFG, which builds on C-PCFGs by incorporating pre-trained representations from LLMs into the C-PCFG inference network. Specifically, we formulate the inference network as:

$$\mathcal{E}(\boldsymbol{x}) = \text{LLM}(\boldsymbol{x}) \tag{1}$$

$$g(\boldsymbol{x}) = \text{FC}(m(\mathcal{E}(\boldsymbol{x}))) \tag{2}$$

where $m$ represents a mean-pool operation. Here, an LLM is used to obtain text embeddings for each sentence $\boldsymbol{x}$, which are then fed to a fully connected (FC) layer as the C-PCFG inference network. We display an example in Figure 3, we feed a sentence $\boldsymbol{x}$ ("*A squirrel jumps on a stump*") into LLMs to obtain sentence embeddings. Then we input the sentence embeddings into a fully-connected layer and obtain the latent $\boldsymbol{z}$. Finally, we feed $\boldsymbol{z}$ into C-PCFG to obtain the targeted constituency parse tree. Our approach does not require the addition of multimodal data for regularization in comparison with previous works. In Section 4, we show how the addition of LLM features can help improve performance of C-PCFG models.

## 4 EXPERIMENTS

### 4.1 IMAGE-ASSISTED PARSING

We compare LC-PCFG against VG-NSL (Shi et al., 2019) and VC-PCFG (Zhao & Titov, 2020), prior unsupervised grammar induction methods that employ visual regularization via visual matching loss.

Table 1: **Grammar induction with image and text**. Corpus-level F1 (C-F1) and sentence-level F1 (S-F1) scores on the MSCOCO 2014 caption dataset. We compare LC-PCFG against simple rule-based baselines (top, from Zhao & Titov (2020)), prior state-of-the-art methods that employ image data (middle), and methods, including ours, that use purely textual data (bottom). *Compound* and *Visual* respectively indicate C-PCFG and VC-PCFG, methods used in prior work. LC-PCFG significantly outperforms all prior work, *without* requiring paired visual data.

| Method | RGB | LLM | Params (M) | C-F1 | S-F1 |
|---|---|---|---|---|---|
| *Rule-based baselines* | | | | | |
| Left Branching | No | No | - | 15.1 | 15.7 |
| Right Branching | No | No | - | 51.0 | 51.8 |
| Random Trees | No | No | - | $24.2_{\pm 0.3}$ | $24.6_{\pm 0.2}$ |
| *Methods using extra-linguistic inputs* | | | | | |
| VG-NSL (Shi et al., 2019) | Yes | No | - | $50.4_{\pm 0.3}$ | - |
| VC-PCFG (Zhao & Titov, 2020) | Yes | No | 41.5 | $59.3_{\pm 8.2}$ | $59.4_{\pm 8.3}$ |
| VC-PCFG++ | Yes | No | 41.5 | $64.2_{\pm 7.0}$ | $64.6_{\pm 7.2}$ |
| *Methods using only textual inputs* | | | | | |
| C-PCFG (Kim et al., 2019) | No | No | 15.3 | $53.6_{\pm 4.7}$ | $53.7_{\pm 4.6}$ |
| **LC-PCFG** (Ours) | No | Yes | **6.2** | $\mathbf{67.2}_{\pm 1.1}$ | $\mathbf{67.8}_{\pm 1.2}$ |

This regularization encourages the learned representations of caption constituents to be aligned with their matching captions. VG-NSL and VC-PCFG require paired image-caption data during training.
**Setup.** We follow the experimental setup of Zhao & Titov (2020), evaluating on their splits of the MSCOCO 2014 dataset (Lin et al., 2014). Because MSCOCO does not provide captions for their test set, a portion of the validation set is used as a held-out test set. Every image in the dataset is associated with 5 captions, and the final dataset consists of 82,783 training, 1,000 validation, and 1,000 test images. During preprocessing, all sentences are lowercased and numbers are replaced with "N". For models using word embedding matrices, the most frequent 10,000 words (based on white-space tokenization) are maintained with all other words mapped to a special UNK token. Captions greater than 45 words in length are removed. For LC-PCFG, we preprocess the dataset by extracting token-level embeddings for each caption from the last layer of an LLM.

**Evaluation.** Because the MS-COCO dataset does not have annotated ground truth parse trees, we follow prior work and use Benepar (Kitaev & Klein, 2018), a supervised neural parser, to generate parse trees for evaluation. Predicted parse trees are compared against generated trees using F1 scores. Due to instabilities observed during training, each method is trained with 10 random seeds and then the mean and standard deviation over the top 4 seeds (based on validation F1) are reported.

**Implementation.** For replicated baseline models, we use the implementation and hyper-parameters provided by Zhao & Titov (2020),[1] which implement the C-PCFG inference network as an LSTM (Hochreiter & Schmidhuber, 1997) encoder on learnt word embeddings, both learned from scratch on the task without pre-training.

VC-PCFG uses a ResNet-152 backbone for embedding images. We modernize the visual backbone to ResNetV1.5 - 152 (ResNetV1.5, 2022) and improve the optimization hyper-parameters (learning rate, network dropout) to obtain VC-PCFG++. VCPCFG++ outperforms VC-PCFG by a substantial margin of about 3 points in both corpus and sentence-level F1 scores.

For LC-PCFG, we use an OPT-2.7B (Zhang et al., 2022b) backbone to extract token-level embeddings for sentences. Embeddings are then mean-pooled and passed through a single linear layer inference network. We use dropout of 0.5 on both the mean-pooled sentence embedding as well as the output latent vector from the inference network.

### 4.1.1 RESULTS

Test F1 scores are shown in Table 1. Our text-only method (LC-PCFG) achieves the highest overall corpus-level F1 (C-F1) and sentence-level F1 (S-F1) scores. Note that LC-PCFG does not use paired

---

[1] https://github.com/zhaoyanpeng/vpcfg

Table 2: **Grammar induction with video and text**. Comparison across three video-text parsing benchmark datasets (DiDeMo, YouCook2 & MSRVTT) against simple rule-based baselines (top), prior state-of-the-art that uses extra-linguistic signals such as video, face and scene cues for grammar induction (middle) and works, including ours (LCPCFG), that learn purely from textual data (bottom). LC-PCFG employs token-level LLM features that replace features extracted from additional modalities, and outperforms all prior works using purely textual data. We also provide the unabridged version presented in the Appendix.

| PCFG Method | LLM | RGB | DiDeMo | | YouCook2 | | MSRVTT | |
|---|---|---|---|---|---|---|---|---|
| | | | C-F1 | S-F1 | C-F1 | S-F1 | C-F1 | S-F1 |
| *Rule-based baselines* | | | | | | | | |
| Left Branching | No | No | 16.2 | 18.5 | 6.8 | 5.9 | 14.4 | 16.8 |
| Right Branching | No | No | 53.6 | 57.5 | 35.0 | 41.6 | 54.2 | 58.6 |
| Random | No | No | $29.4_{\pm 0.3}$ | $32.7_{\pm 0.5}$ | $21.2_{\pm 0.2}$ | $24.0_{\pm 0.2}$ | $27.2_{\pm 0.1}$ | $30.5_{\pm 0.1}$ |
| *Methods using extra-linguistic inputs* | | | | | | | | |
| VC-PCFG (Zhao & Titov, 2020) | No | Yes | $42.2_{\pm 12.3}$ | $43.2_{\pm 14.2}$ | $42.3_{\pm 5.7}$ | $47.0_{\pm 5.6}$ | $49.8_{\pm 4.1}$ | $54.2_{\pm 4.0}$ |
| MMC-PCFG (Zhang et al., 2021) | No | Yes | $55.0_{\pm 3.7}$ | $58.9_{\pm 3.4}$ | $44.7_{\pm 5.2}$ | $48.9_{\pm 5.7}$ | $56.0_{\pm 1.4}$ | $60.0_{\pm 1.2}$ |
| *Methods using only textual inputs* | | | | | | | | |
| C-PCFG (Kim et al., 2019) | No | No | $38.2_{\pm 5.0}$ | $40.4_{\pm 4.1}$ | $37.8_{\pm 6.7}$ | $41.4_{\pm 6.6}$ | $50.7_{\pm 3.2}$ | $55.0_{\pm 3.2}$ |
| **LC-PCFG** (Ours) | Yes | No | $\mathbf{57.1}_{\pm 4.7}$ | $\mathbf{60.0}_{\pm 5.2}$ | $\mathbf{52.4}_{\pm 0.1}$ | $\mathbf{57.7}_{\pm 0.1}$ | $\mathbf{56.1}_{\pm 3.6}$ | $\mathbf{61.2}_{\pm 3.7}$ |

visual features, and contains 85% fewer parameters compared to previous state-of-the-art method (VC-PCFG).

## 4.2 VIDEO-ASSISTED PARSING

We find that LLMs help unsupervised parsing in comparison with image-regularized models. However, images are a static snapshot of the world, and often fail to reflect constituents such as verb phrases (e.g., temporal actions). Prior work presented MultiModal Compound PCFG (MMC-PCFG) to investigate the benefit of leveraging multiple modalities found in videos (Zhang et al., 2021), such as audio and action features. MMC-PCFG aggregates these multimodal features for unsupervised grammar induction and achieves a substantial improvement over previous methods. To better understand grounding in perception and action, we compare our approach with state-of-the-art methods such as MMC-PCFG which use multimodal information for parsing tasks in video settings.

**Setup.** Following MMC-PCFG, we use three benchmarking video datasets for our experiments: Distinct Describable Moments (*DiDeMo*) (Anne Hendricks et al., 2017), Youtube Cooking (*YouCook2*) (Zhou et al., 2018) and MSRVideo to Text (*MSRVTT*) (Xu et al., 2016). DiDeMo consists of unedited, personal videos in diverse visual settings with pairs of localized video segments and referring expressions. It includes 32994, 4180 and 4021 video-sentence pairs in the training, validation, and test sets. YouCook2 contains 2000 videos that are nearly equally distributed over 89 recipes. Each video contains 3–16 procedure segments. It includes 8713, 969 and 3310 video-sentence pairs in the training, validation and test sets. MSRVTT is a large-scale benchmark for video understanding with 10K web video clips with 41.2 hours and 200K clip-sentence pairs in total. It includes 130260, 9940 and 59794 video-sentence pairs across all the data splits.

The extracted multimodal features (Zhang et al., 2021) includes object features (SENet (Xie et al., 2017)), action features (I3D (Carreira & Zisserman, 2017)), scene (Huang et al., 2017; Zhou et al., 2017), audio (Hershey et al., 2017), OCR (Deng et al., 2018; Liu et al., 2018), face (Liu et al., 2016; He et al., 2016) and speech (Mikolov et al., 2013).

We run all experiments 4 times, 10 epochs each, with different random seeds and report the mean & standard deviation on C-F1 and S-F1.

### 4.2.1 RESULTS

We compare our results with C-PCFG, VC-PCFG (which contains a single extra-linguistic modality), and MMC-PCFG (which contains multiple extra-linguistic modalities) (Table 2). We find that LC-PCFG outperforms the parsing performance of video-regularized models for all three benchmark datasets. For example, MMC-PCFG learns all modalities together and achieves 44.7 as the average

Table 3: **Transferring Learnt Grammar**. Models are trained on the 'Trainset' data and evaluated without additional training on the target benchmarks (DiDeMo, YouCook2 & MSRVTT) on the Sentence-level F1 (S-F1) and Corpus-level F1 (C-F1) metrics. All HowTo100M results are reported on 592k samples.

| Method | Trainset | DiDeMo | | YouCook2 | | MSRVTT | |
|---|---|---|---|---|---|---|---|
| | | C-F1 | S-F1 | C-F1 | S-F1 | C-F1 | S-F1 |
| MMC-PCFG | DiDeMo | $55.0_{+3.7}$ | $58.9_{+3.4}$ | $49.1_{+4.4}$ | $53.0_{+4.9}$ | $49.6_{+1.4}$ | $53.8_{+0.9}$ |
| MMC-PCFG | YouCook2 | $40.1_{+4.4}$ | $44.2_{+4.4}$ | $44.7_{+5.2}$ | $48.9_{+5.7}$ | $34.0_{+6.4}$ | $37.5_{+6.8}$ |
| MMC-PCFG | MSRVTT | $59.4_{+2.9}$ | $62.7_{+3.3}$ | $49.6_{+3.9}$ | $54.2_{+4.1}$ | $56.0_{+1.4}$ | $60.0_{+1.2}$ |
| MMC-PCFG | HowTo100M | $58.5_{+7.3}$ | $62.4_{+7.9}$ | $53.9_{+6.6}$ | $58.0_{+7.1}$ | $55.1_{+7.0}$ | $60.2_{+8.0}$ |
| PTC-PCFG | HowTo100M | $\mathbf{61.3}_{+3.9}$ | $\mathbf{65.2}_{+5.3}$ | $58.9_{+2.5}$ | $63.2_{+2.3}$ | $57.4_{+4.6}$ | $62.8_{+5.7}$ |
| **LC-PCFG** (Ours) | HowTo100M | $60.6_{+5.2}$ | $61.5_{+6.1}$ | $\mathbf{61.1}_{+2.1}$ | $\mathbf{65.2}_{+1.4}$ | $\mathbf{59.4}_{+5.0}$ | $\mathbf{63.0}_{+5.8}$ |

C-F1 score on the YouCook2 dataset, which displays obvious improvement over previous methods. However, our approach achieves $52.4$ as the average C-F1 score, a substantial improvement of 7.7 points, all without using any multimodal data.

### 4.3 LARGE-SCALE VIDEO PRETRAINING

Recently Zhang et al. (2022a) proposed Pre-Trained Compound PCFGs (PTC-PCFG), a simple model that can better capture video-text matching information and obtain substantially higher performance on benchmark datasets by training on millions of out-of-domain instructional, noisy YouTube videos. PTC-PCFG shows that learning from large-scale video data enables stronger and more robust performance, even compared to models trained on in-domain data.

Given such a situation, our method only trains with the captions from the same data from *HowTo100M* (Miech et al., 2019), without any additional multi-modality knowledge. HowTo100M (HT) is a large-scale dataset of 136 million video clips sourced from 1.22M narrated instructional web videos depicting humans performing and describing over 23k different visual tasks. Following the settings in PTC-PCFG, we train the model using the same 592k training samples. Instead of using all the media resources, we only train with sentences.

We report the comparison and robustness of models in Table 3. It could be noticed that our approach still yields superior scores without finetuning the model with any benchmark dataset. For example, when training with YouCook2, MMC-PCFG could only obtain $44.7_{+5.2}$ for C-F1 score and $48.9_{+5.7}$ for S-F1 score. Instead, when training with HT (592k) dataset, PTC-PCFG could achieve $58.9_{+2.5}$ for C-F1 and $63.2_{+2.3}$ for S-F1. This demonstrates the effectiveness of the PTC-PCFG model by training with a large amount of data. However, our approach only trains with the sentences from HT (592k) dataset and obtains superior results: $61.1_{+2.1}$ for C-F1 score and $65.2_{+1.4}$ for S-F1 score, suggesting that the video signal may be redundant when using LLM embeddings.

**Runtime.** Since our approach doesn't require multimodal-text matching, it requires much less computation than MMC-PCFG and PTC-PCFG. To justify this point, we follow the setting of PTC-PCFG and calculate the approximate time cost for each run of different models using 2 V100 GPUs. We report running time comparisons in Table 4. We note that our approach runs much faster in comparison with other state-of-the-art methods.

## 5 MODEL ANALYSIS

### 5.1 PERPLEXITY-BASED EVALUATION

In order to facilitate comparisons with prior works, in Section 4 we follow the model selection procedure used in Zhao & Titov (2020); Zhang et al. (2021; 2022a) of selecting runs for test evaluation based on validation set C-F1 score. However, this model selection procedure assumes gold parse trees are available during validation. To validate the conclusions obtained from this model selection procedure while relaxing the assumption of gold trees being available at validation time, we repeat our experiments but instead use perplexity (PPL) (Chen et al., 1998) to perform model

Table 4: **Training Time Evaluation** for both image-based (top) and video-based (bottom) grammar induction methods. Run-time for both extracting the embeddings ('Embedding') and total training ('Training') are reported. Image and Video results are benchmarked on a single 12G 2080 Ti and on $2\times$ 32G V100 respectively.

| PCFG Method | Embedding (hours) | Training (hours) |
|---|---|---|
| C-PCFG | - | 7.6 |
| VC-PCFG | 0.25 | 13.3 |
| LC-PCFG | | 8.0 |
| C-PCFG | - | 1.5 |
| MMC-PCFG | >25 | 15 |
| PTC-PCFG | >25 | 10 |
| LC-PCFG (Ours) | 2.5 | 1.7 |

Table 5: **Unsupervised Run Selection Criterion for Unsupervised Grammar Induction**. Corpus-level F1 scores using validation set F1 ('Val-F1'), perplexity ('PPL'), and mean branching factor ('MBF', the average proportion between leaves in the right and left branches of nodes in each tree across the corpus). Unlike Validation-F1 based-selection, PPL and MBF do not require gold trees during validation.

| PCFG Method | Run Selection Criteria | | |
|---|---|---|---|
| | Val-F1 | PPL | MBF |
| C-PCFG | $60.1_{\pm 4.6}$ | $52.0_{\pm 7.5}$ | $56.8_{\pm 9.3}$ |
| VC-PCFG | $61.3_{\pm 2.6}$ | $55.3_{\pm 10.2}$ | $51.0_{\pm 13.4}$ |
| **LC-PCFG** (Ours) | $\mathbf{67.2}_{\pm 1.1}$ | $\mathbf{67.2}_{\pm 1.1}$ | $\mathbf{65.3}_{\pm 2.1}$ |

Table 6: **Unsupervised Run Selection Criterion for Unsupervised Grammar Induction**. Similar to Table 5, we report the results of run selection based on validation perplexity (PPL) for video benchmarks (Section 5.1).

| PCFG Method | DiDeMo | | YouCook2 | | MSRVTT | |
|---|---|---|---|---|---|---|
| | C-F1 | S-F1 | C-F1 | S-F1 | C-F1 | S-F1 |
| Compound Kim et al. (2019) | $40.4_{\pm 10.1}$ | $42.1_{\pm 9.1}$ | $38.6_{\pm 7.2}$ | $42.8_{\pm 7.7}$ | $49.2_{\pm 3.8}$ | $53.1_{\pm 4.0}$ |
| Multi-modal Zhang et al. (2021) | $42.1_{\pm 12.6}$ | $45.7_{\pm 12.4}$ | $38.9_{\pm 3.6}$ | $43.8_{\pm 3.3}$ | $48.1_{\pm 1.0}$ | $52.4_{\pm 0.9}$ |
| **LC-PCFG** (Ours) | $\mathbf{46.3}_{\pm 6.9}$ | $\mathbf{49.9}_{\pm 7.3}$ | $\mathbf{46.7}_{\pm 1.1}$ | $\mathbf{52.4}_{\pm 0.8}$ | $\mathbf{50.5}_{\pm 4.0}$ | $\mathbf{55.2}_{\pm 4.4}$ |

selection:

$$PPL(X) = -\frac{1}{t}\sum_{i}^{t} log\ p(x_i|x_{<i})$$

where $X = (x_1, x_2, x_3, ..., x_t)$ is a tokenized sequence of words and $p(x_i|x_{<i})$ represents the log-likelihood of the ith token conditioned on the preceding tokens $x_{<i}$.

PPL allows us to perform model selection without relying on supervised examples needed for computing metrics such as C-F1 or S-F1. We train models with 10 random seeds, and then use PPL to select the four best-performing seeds.

Test C-F1 performance on image-assisted parsing is shown in Table 5. Under perplexity-based model selection, LC-PCFG also consistently outperforms methods that use extra-linguistic inputs, validating the conclusions from Section 4. We also report the result in Table 6 on video-assisted parsing, where we notice the same conclusion.

## 5.2 PARSER BRANCHING

Given that English is a right-branching language, we measure the predilection of learned parsers to lean one way or the other. Specifically, for each branch in each parse tree we measure the proportion of leaves under the right branch over those of the left branch. This proportion is then averaged across all nodes in the tree to produce an average score $s$, which we refer to as the branching factor of the tree ($s > 1.0$ implies right-branching, $s < 1.0$ is left-branching). Formally, for each parse tree $t$ with $|t|$ nodes $n \in t$ we compute the mean over nodes' ratio of leaves in their right and left branches:

$$\text{MBF}(t) = \frac{1}{|t|} \sum_{n \in t} \frac{\text{CR}(n)}{\text{CL}(n)}$$

where CR and CL are the respective counts of leaves under the right and left branches of a node.

Table 7: MBF for each method on image-assisted parsing. The lower the better.

| PCFG Method | MBF |
|---|---|
| C-PCFG | $3.4_{\pm 0.3}$ |
| VC-PCFG | $3.4_{\pm 0.3}$ |
| LC-PCFG | $2.5_{\pm 0.7}$ |

Table 7 presents the mean branching factor (MBF) for each model (computed over 10 seeds). We find that all evaluated parsers are right-branching. Further, in Table 5, we show that MBF is generally negatively correlated with parsing performance (with LC-PCFG having the greatest negative correlation), suggesting that selecting for more balanced branching factors may also be used as an unsupervised seed selection method.

### 5.3 MODEL ABLATIONS

We perform a series of ablations on parsers trained on the MSCOCO dataset to understand the effect of different model components, such as the latent variable $z$, on the learned parsing ability.

To understand the contribution of the latent variable $z$, we perform several $z$ ablations, in both training and evaluation (results are shown in Table 8). First is the set of inference-time ablations, where only in inference, we zero out the latent variable, $z$ ('Zero-$z$'), or randomly shuffle $z$ within an evaluation batch ('Random-$z$'). We also perform a second set of training-time ablations by training a C-PCFG model without latents ('Zero-Train', a vanilla neural PCFG model). We find that inference-time ablations on the latent yield comparable performance to the default parsers, whereas omitting the latent during training yields reduced performance from the standard C-PCFG/LC-PCFG models. These results suggest that the latent variable

Table 8: **Parser Ablations.** Corpus-level F1 scores for PCFG parsers under ablations. We compare the **Default** formulations to conditions zeroing out the latent $z$ ('Zero-$z$'), randomly shuffling latents across a batch ('Random-$z$'), shuffling words in each caption ('Shuffle') or zeroing captions out ('Zero-C'), as well as zeroing out latents during training ('Zero-Train'). Note that C-PCFG and LC-PCFG are functionally equivalent in the Zero-Train condition because LLM features are only used in latent computation.

| Ablation | Test Corpus F1 | | |
|---|---|---|---|
| | C-PCFG | VC-PCFG | LC-PCFG |
| Default | $60.1_{\pm 4.6}$ | $61.3_{\pm 2.6}$ | $67.2_{\pm 1.1}$ |
| Zero-$z$ | $60.3_{\pm 5.2}$ | $60.6_{\pm 2.6}$ | $67.2_{\pm 1.1}$ |
| Random-$z$ | $60.3_{\pm 5.2}$ | $60.9_{\pm 2.5}$ | $67.2_{\pm 1.1}$ |
| Shuffle | $30.0_{\pm 0.7}$ | $31.0_{\pm 0.9}$ | $40.6_{\pm 1.0}$ |
| Zero-C | $35.2_{\pm 16.1}$ | $44.6_{\pm 7.6}$ | $48.6_{\pm 7.5}$ |
| Zero-Train | $57.1_{\pm 6.5}$ | $58.8_{\pm 0.9}$ | $57.1_{\pm 6.5}$ |

may be largely ignored at inference time, but that it serves an important role in the learning process of the parser.

Lastly, we also ablate the input sentences. We evaluate parsers when shuffling ('Shuffle') or zeroing out input caption embeddings ('Zero-C', word embeddings for VC-PCFG or LLM embeddings for LC-PCFG). We find that ablating the input sentences substantially reduces test performance, suggesting that learned parsers do not merely degenerate to a learned prior at inference time.

## 6 CONCLUSION AND FUTURE WORK

**Conclusion.** In this work, we propose a strong vision-free baseline LC-PCFG for grammar induction. We compare the performance of several prior multimodal models to LC-PCFG and observe that LC-PCFG performs generally better than previous multimodal models, indicating that text-only models such as LLMs may contain visually grounded signals for unsupervised grammar induction.

**Future Work.** Given that this study focuses on unsupervised grammar induction, there remains a performance disparity compared to supervised techniques. For future endeavors, we may explore the integration of multimodal signals into our vision-free baseline as a means to bridge the gap and boost the model's performance.

## ETHICS STATEMENT

Since our work deals more broadly with the development of LLMs and multi-modal methods for grammar induction, we do not anticipate direct ethical harm from this specific NLP task. Our experiments suggest that our vision-free baseline could generally outperform computationally intensive multimodal approaches for grammar induction. We aim for our results to encourage the community to explore less computationally demanding methods.

## REPRODUCIBILITY STATEMENT.

In this paper, we mainly apply and evaluate our method on open-sourced models C-PCFG, VC-PCFG, MMC-PCFG, PTC-PCFG, which allows the reproduction of results in this work. We commit to releasing our code to inspire further research in this domain.

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
