# A Vision-free baseline for Multimodal Grammar Induction (Supplementary Materials)

## 1 Video-assisted Parsing

In Table 1 we present the unabridged version of comparisons for grammar induction with video and text, corresponding to Table 2 in the paper.

Table 1: Comparison with multi-modalities on three benchmark datasets. Note that our method, using features from OPT-2.7B, yields superior results despite not being regularized by multiple modalities. "W/ MD" denotes using multimodal data.

| | Method | W/ MD | DiDeMo | | | | | YouCook2 | | MSRVTT | |
|---|---|---|---|---|---|---|---|---|---|---|---|
| | | | NP | VP | PP | C-F1 | S-F1 | C-F1 | S-F1 | C-F1 | S-F1 |
| | LBranch | ✗ | 41.7 | 0.1 | 0.1 | 16.2 | 18.5 | 6.8 | 5.9 | 14.4 | 16.8 |
| | RBranch | ✗ | 32.8 | **91.5** | 66.5 | 53.6 | 57.5 | 35.0 | 41.6 | 54.2 | 58.6 |
| | Random | ✗ | $36.5_{\pm 0.6}$ | $30.5_{\pm 0.5}$ | $30.1_{\pm 0.5}$ | $29.4_{\pm 0.3}$ | $32.7_{\pm 0.5}$ | $21.2_{\pm 0.2}$ | $24.0_{\pm 0.2}$ | $27.2_{\pm 0.1}$ | $30.5_{\pm 0.1}$ |
| | C-PCFG | ✗ | $\mathbf{72.9}_{\pm 5.5}$ | $16.5_{\pm 6.2}$ | $23.4_{\pm 16.9}$ | $38.2_{\pm 5.0}$ | $40.4_{\pm 4.1}$ | $37.8_{\pm 6.7}$ | $41.4_{\pm 6.6}$ | $50.7_{\pm 3.2}$ | $55.0_{\pm 3.2}$ |
| VC-PCFG | Object | ✓ | $70.5_{\pm 15.3}$ | $25.7_{\pm 15.9}$ | $36.5_{\pm 24.6}$ | $42.6_{\pm 10.4}$ | $44.0_{\pm 10.4}$ | $39.9_{\pm 8.7}$ | $44.9_{\pm 8.3}$ | $52.2_{\pm 1.2}$ | $56.0_{\pm 1.6}$ |
| | Action | ✓ | $57.9_{\pm 13.5}$ | $45.7_{\pm 14.1}$ | $45.8_{\pm 17.2}$ | $45.1_{\pm 6.0}$ | $49.2_{\pm 6.0}$ | $40.6_{\pm 3.6}$ | $45.7_{\pm 3.2}$ | $54.5_{\pm 1.6}$ | $59.1_{\pm 1.7}$ |
| | R2P1D | ✓ | $61.2_{\pm 8.5}$ | $38.1_{\pm 5.4}$ | $62.1_{\pm 4.1}$ | $48.1_{\pm 4.4}$ | $50.7_{\pm 4.2}$ | $39.4_{\pm 8.1}$ | $44.4_{\pm 8.3}$ | $54.0_{\pm 2.5}$ | $58.0_{\pm 2.3}$ |
| | S3DG | ✓ | $61.3_{\pm 13.4}$ | $31.7_{\pm 16.7}$ | $51.8_{\pm 8.0}$ | $44.0_{\pm 2.7}$ | $46.5_{\pm 5.1}$ | $39.3_{\pm 6.5}$ | $44.1_{\pm 6.6}$ | $50.7_{\pm 3.2}$ | $54.7_{\pm 2.9}$ |
| | Scene | ✓ | $62.2_{\pm 9.6}$ | $30.6_{\pm 12.3}$ | $41.1_{\pm 24.8}$ | $41.7_{\pm 6.5}$ | $44.9_{\pm 7.4}$ | $-$ | $-$ | $54.6_{\pm 1.5}$ | $58.4_{\pm 1.3}$ |
| | Audio | ✓ | $64.2_{\pm 18.6}$ | $21.3_{\pm 26.5}$ | $34.7_{\pm 11.0}$ | $38.7_{\pm 3.7}$ | $39.5_{\pm 5.2}$ | $39.2_{\pm 4.7}$ | $43.3_{\pm 4.9}$ | $52.8_{\pm 1.3}$ | $56.7_{\pm 1.4}$ |
| | OCR | ✓ | $64.4_{\pm 15.0}$ | $27.4_{\pm 19.5}$ | $42.8_{\pm 31.2}$ | $41.9_{\pm 16.9}$ | $44.6_{\pm 17.5}$ | $38.6_{\pm 5.5}$ | $43.2_{\pm 5.6}$ | $51.0_{\pm 3.0}$ | $55.5_{\pm 3.0}$ |
| | Face | ✓ | $60.8_{\pm 16.0}$ | $31.5_{\pm 17.0}$ | $52.8_{\pm 9.8}$ | $43.9_{\pm 4.5}$ | $46.3_{\pm 5.5}$ | $-$ | $-$ | $50.5_{\pm 2.6}$ | $54.5_{\pm 2.6}$ |
| | Speech | ✓ | $61.8_{\pm 12.8}$ | $26.6_{\pm 17.6}$ | $43.8_{\pm 34.5}$ | $40.9_{\pm 16.0}$ | $43.1_{\pm 16.1}$ | $-$ | $-$ | $51.7_{\pm 2.6}$ | $56.2_{\pm 2.5}$ |
| | Concat | ✓ | $68.6_{\pm 8.6}$ | $24.9_{\pm 19.9}$ | $39.7_{\pm 19.5}$ | $42.2_{\pm 12.3}$ | $43.2_{\pm 14.2}$ | $42.3_{\pm 5.7}$ | $47.0_{\pm 5.6}$ | $49.8_{\pm 4.1}$ | $54.2_{\pm 4.0}$ |
| | MMC-PCFG | ✓ | $67.9_{\pm 9.8}$ | $52.3_{\pm 9.0}$ | $63.5_{\pm 8.6}$ | $55.0_{\pm 3.7}$ | $58.9_{\pm 3.4}$ | $44.7_{\pm 5.2}$ | $48.9_{\pm 5.7}$ | $56.0_{\pm 1.4}$ | $60.0_{\pm 1.2}$ |
| | **LC-PCFG** | ✗ | $71.1_{\pm 6.6}$ | $47.4_{\pm 12.6}$ | $\mathbf{76.9}_{\pm 7.3}$ | $\mathbf{57.1}_{\pm 4.7}$ | $\mathbf{60.0}_{\pm 5.2}$ | $\mathbf{52.4}_{\pm 0.1}$ | $\mathbf{57.7}_{\pm 0.1}$ | $\mathbf{56.1}_{\pm 3.6}$ | $\mathbf{61.2}_{\pm 3.7}$ |