# OpenReview forum: "A Vision-free Baseline for Multimodal Grammar Induction"
_ICLR.cc/2024/Conference — ICLR 2024 Conference Withdrawn Submission_

### Official Review · Reviewer_gdiQ · 2023-10-29

**Soundness:** 2 fair
**Presentation:** 3 good
**Contribution:** 2 fair
**Rating:** 3
**Confidence:** 5

**Summary:**

The paper discusses the exploration of using large language models (LLMs) trained only with text to improve grammar induction in multimodal datasets. The approach, called LLM-based C-PCFG (LC-PCFG), outperforms previous multimodal methods and achieves state-of-the-art performance in grammar induction for various multimodal datasets. Compared to image-aided grammar induction, LC-PCFG shows remarkable improvement, with significantly fewer parameters and faster training speed. In three video-assisted grammar induction benchmarks, LC-PCFG outperforms the prior state-of-the-art, with 8.8× faster training. These findings suggest that text-only language models may contain visually grounded cues that assist in grammar induction in multimodal contexts. The results also highlight the importance of establishing a strong vision-free baseline for evaluating the benefits of multimodal approaches.

**Strengths:**

- The paper proposed to use powerful LLM-based embeddings to improve multimodal grammar induction.
- The proposed LC-PCFG outperforms the prior state-of-the-art methods.
- The experiments are sufficient. The writing and presentation are clear and easy to read.

**Weaknesses:**

-	The title (“vision-free”, “multimodal”) is confusing because the proposed LC-PCFG requires only textual inputs.
-	The major concern is that the proposed method is based on a pre-trained model, which imports billions of extra training data. It’s not fair to compare with the method without pre-training. For example, the original implementation of C-PCFG takes a CNN-based network for text embedding and the version used in this paper takes a LSTM network, both learned from scratch on the task without pre-training. If the author could provide the results of a reproduced version with pretrained embeddings, the improvement will be more convincing.
-	I’m not sure if the using of pretraining model will weaken the point on "unsupervised" grammar induction (because the pre-training language model now may require amount of corpus and multi-task objectives). Could the author explain this issue.
-	I guess the “#Param” column in Table 1 should be “#Training Param”. The pre-training parameters (OPT-7B) are not included. The same problem in Table 4, i.e. the embedding and training hours do not include the pre-training stage.
-	Could the author provide the performance of each type of label (NP, VP…)?
-	To sum up, it seems that the improvement totally comes from the LLM-based embeddings. My concern is that if the LLM already has powerful ability such as understanding syntax knowledge and even high-level semantic knowledge, why we use it on a very basic task. Therefore, research motivation should be clarified.
-	Limitations should be discussed somewhere.

**Questions:**

see weakness.

---

### Official Review · Reviewer_5YMo · 2023-10-31

**Soundness:** 1 poor
**Presentation:** 3 good
**Contribution:** 2 fair
**Rating:** 1
**Confidence:** 4

**Summary:**

This paper investigates the problem of compound PCFG (C-PCFG; Kim et al., 2019) induction with features from large language models (LLMs), and claims such an approach could serve as a vision-free baseline for multimodal grammar induction models.

**Strengths:**

The presentation is very clear, and the paper is very easy to understand.

**Weaknesses:**

The motivation for this work does not make any sense to me for the following three reasons.

1. The comparison between the LLM-based approach and existing multimodal grammar induction work is unfair regarding data availability and cognitive plausibility.
The primary goal of the multimodal grammar induction work, critiqued in this paper, is not leaderboard chasing.
The multimodal setting provides a simplified environment to model the syntax acquisition process of humans, and the amount of training data should be strictly limited to what humans can access in the language acquisition processes.
No one acquires their first language by solely reading trillions of words and predicting the next token, like how LLMs are trained.
The comparison between an LLM-based C-PCFG and multimodal grammar induction is unfair regarding data exposure.

2. There may be data contamination issues in LLM training.
The training data of LLMs are not curated to exclude constituency parse trees.
That is, although they are not explicitly trained with the objectives used in supervised constituency parsing, they are exposed to the specific supervision and, therefore, should not be considered as fully unsupervised grammar *induction*.
As an alternative, I suggest the authors try prompting LLMs (such as ChatGPT or LLaMa-2) using the following templates:

    > Q: In PTB style, what is the constituency parse tree of the following sentence? \
    [SENTENCE] \
    A:

    With some simple post-hoc checking (e.g., ensuring the output sentence is faithful to the input), this approach will hopefully result in even better results reported in this paper.
    Does this mean LLMs successfully *induce* the grammar? The answer is absolutely no. They are learning from explicit supervision in the training set.

3. I can't see why the authors specifically target multimodal grammar induction and not compare it to text-based grammar induction results.
As a background, DIORA (Drozdov et al., NAACL 2019; not even cited in this paper as a representative recent work in grammar induction) used ELMo, one of the first-generation "LLMs," for the initialization of word embeddings.
I suggest the authors check out DIORA and its follow-up work and re-consider the claim.

In addition to motivation, recent work on text-only grammar induction is almost entirely missing.
Please at least check out the following ones, and I suggest the authors to do a more comprehensive literature research using these as starting points:
- Drozdov et al. Unsupervised latent tree induction with deep inside-outside recursive autoencoders. NAACL 2019
- Drozdov et al. Unsupervised parsing with S-DIORA: Single tree encoding for deep inside-outside recursive autoencoders. EMNLP 2020
- Jin et al. Variance of average surprisal: a better predictor for quality of grammar from unsupervised PCFG induction. ACL 2019
- Kim et al. Unsupervised Recurrent Neural Network Grammars. NAACL 2019

**Questions:**

My only questions are about the motivation of this work (details above). It would be great if the authors could justify their motivation by addressing the comments raised above.

---

### Official Review · Reviewer_qYkX · 2023-11-01

**Soundness:** 1 poor
**Presentation:** 2 fair
**Contribution:** 1 poor
**Rating:** 3
**Confidence:** 2

**Summary:**

This work proposes an unsupervised text parsing with vision information using large language model without any vision information. Experimental results comparing other vision aware models, the proposed method achieve better results by simply leveraging representations from a large language model using compound probabilistic context free grammars.

**Strengths:**

The work presents systematic comparisons with vision aware unsupervised parsing and demonstrates the gains of the proposed approach.

**Weaknesses:**

* Vision-free unsupervised text parsing for a vision aware task is merely unsupervised text parsing. Thus, this work should compare with various latest unsupervised methods for a fair comparison with and without the use of the representations from large language models:

  - Yikang Shen, Shawn Tan, Alessandro Sordoni, and Aaron Courville. Ordered neurons: Integrating tree structures into recurrent neural networks. In ICLR 2019.
  - Andrew Drozdov, Patrick Verga, Mohit Yadav, Mohit Iyyer, and Andrew McCallum. Unsupervised latent tree induction with deep inside-outside recursive auto-encoders. In NAACL-HLT 2019.
  - Steven Cao, Nikita Kitaev, and Dan Klein. Unsupervised parsing via constituency tests. In EMNLP 2020.
  - Jiaxi Li and Wei Lu. Contextual distortion reveals constituency: Masked language models are implicit parsers. In ACL 2023.

* This work should investigate vision aware large language models, e.g., mPLUG-Owl, as a comparison if the major focus is a baseline for vision aware unsupervised parsing.

**Questions:**

See the weakness above. I'd rather like to know if the authors run experiments comparing various unsupervised parsing using only textual information and vision aware language models.

---

### Author Response · Authors · 2023-11-17

We thank the authors for their time and their constructive criticism of our submission. Many of the reviewers’ concerns arose from an unclear communication of the motivation for our work (i.e., the exact type of baseline we are proposing, why we do not compare to other text-based grammar induction methods, and why we compare an LLM-trained on huge amounts of data to current multimodal grammar induction words). We have decided that our submission warrants more clarification than would be reasonable in the scope of the author's response. Thus, we have decided to withdraw our submission and would like to reiterate our appreciation for the reviewers’ time and consideration.